# Prediction of Multiple Degenerative Diseases Based on DNA Methylation in a Co-Physiology Mechanisms Perspective

**DOI:** 10.3390/ijms25179514

**Published:** 2024-09-01

**Authors:** Li Zhang, Ruirui Cai, Chencai Wang, Jialong Liu, Zhejun Kuang, Han Wang

**Affiliations:** 1College of Computer Science and Engineering, Changchun University of Technology, Changchun 130051, China; lizhang@ccut.edu.cn (L.Z.); 20212304@stu.ccut.edu.cn (C.W.); liujialong622@gmail.com (J.L.); 2School of Information Science and Technology, Institute of Computational Biology, Northeast Normal University, Changchun 130117, China; cairr097@nenu.edu.cn; 3School of Cyber Security, School of Computer Science and Technology, Changchun University, Changchun 130022, China

**Keywords:** DNA methylation, epigenetic clock, degenerative diseases, deep learning

## Abstract

Degenerative diseases oftentimes occur within the continuous process of aging, and the corresponding clinical manifestations may be neurodegeneration, neoplastic diseases, or various human complex diseases. DNA methylation provides the opportunity to explore aging and degenerative diseases as epigenetic traits. It has already been applied to age prediction and disease diagnosis. It has been shown that various degenerative diseases share co-physiology mechanisms with each other, clues of which may be gained from studying the aging process. Here, we endeavor to predict the risk of degenerative diseases in an aging-relevant comorbid mechanism perspective. Firstly, an epigenetic clock method was implemented based on a multi-scale convolutional neural network, and a Shapley feature attribution analysis was applied to discover the aging-related CpG sites. Then, these sites were further screened to a smaller subset composed of 196 sites by using biomics analysis according to their biological functions and mechanisms. Finally, we constructed a multilayer perceptron (MLP)-based degenerative disease risk prediction model, Mlp-DDR, which was well trained and tested to accurately classify nine degenerative diseases. Recent studies also suggest that DNA methylation plays a significant role in conditions like osteoporosis and osteoarthritis, broadening the potential applications of our model. This approach significantly advances the ability to understand degenerative diseases and represents a substantial shift from traditional diagnostic methods. Despite the promising results, limitations regarding model complexity and dataset diversity suggest directions for future research, including the development of tissue-specific epigenetic clocks and the inclusion of a wider range of diseases.

## 1. Introduction

In current aging populations, the prevalence of degenerative diseases such as osteoarthritis, cardiovascular diseases, and Alzheimer’s disease is rising, presenting significant social and economic challenges [1,2,3]. Researchers have made significant progress in understanding degenerative diseases, but there is still much to be learned regarding their precise causations and mechanisms. Recent studies have highlighted DNA methylation, an epigenetic modification, as playing a crucial role in cellular aging, potentially serving as a novel biomarker for these conditions [4]. Epigenetic clocks, which utilize DNA methylation patterns, offer a method to measure biological age and identify key methylation sites associated with aging. These insights are invaluable for understanding the mechanisms behind degenerative diseases.

Many previous efforts have been undertaken based on DNA methylation analysis, where primarily sodium bisulfite sequencing was widely used to identify DNA methylations by converting unmethylated cytosine to uracil for detection via sequencing. Correspondingly, bisulfite pyrosequencing was used to analyze DNA methylation at two ataxia telangiectasia mutated (ATM) gene loci in leukocytes, suggesting that ATM methylation could be a marker for breast cancer risk [5]. Whole-genome bisulfite sequencing was employed to assess genome-wide DNA methylation in peripheral blood samples, concluding that hypomethylation in pre-diagnosis samples could predict breast cancer risk [6]. Despite their utility, these methods are costly, time consuming, and lack precision in predicting disease risk, underscoring the need for more effective techniques to utilize DNA methylation in diagnosing degenerative diseases.

Recent studies have further emphasized the role of DNA methylation in bone and joint diseases. For example, the m6A modification has been shown to regulate bone homeostasis, offering new potential approaches for osteoporosis treatment [7]. In osteoarthritis, genome-wide association studies (GWAS) and multi-omic analyses have identified key loci associated with disease progression, highlighting the significance of DNA methylation in regulating joint diseases [8]. These findings underscore the broader relevance of DNA methylation beyond traditional age-related diseases and suggest that a more comprehensive approach to modeling these epigenetic changes could enhance diagnostic capabilities across a wider range of conditions.

As computational capabilities progress and epigenetic data grow, the precision of disease risk prediction has improved significantly. Machine learning, particularly through the development of epigenetic clocks [9,10], excels at handling complex clinical data. These models are designed to accurately predict biological age across various tissues and have been a focal point in optimizing structures to refine age acceleration estimates. Horvath’s multi-tissue clock [11], for instance, utilized an Elastic Net regression model to achieve high precision, although it has not yet been directly applied to specific age-related diseases. The Skin and Blood clock [12] and Zhang’s predictors [13] have advanced our understanding of cellular aging processes [14,15]. However, these models often overlook critical interactions among methylation sites that could be crucial for identifying biomarkers for degenerative diseases. Despite ongoing research into the methylation status of disease-associated genes, the focus has predominantly been on cancer, with minimal attention given to degenerative conditions. There is an urgent need for models that integrate a broader range of biological effects and specifically address degenerative diseases to enhance the diagnostic potential of DNA methylation.

The advent of big data has shifted the focus from traditional machine learning to deep learning, owing to its superior capability to model high-dimensional and structurally complex data [16]. Deep learning algorithms excel at automatically extracting rich and abstract features from raw data, offering significant advantages for handling large datasets. Recent studies demonstrate the efficacy of deep neural networks (DNNs) and convolutional neural networks (CNNs) in predicting medical conditions like pediatric stenosis, concussions, and various cancers using DNA methylation data [17,18,19]. These models outperform traditional machine learning techniques like random forests and support vector machines by integrating multiple approaches for enhanced prediction accuracy. While deep learning models have shown promise, ongoing research focuses on enhancing their interpretability and understanding the biological significance of DNA methylation sites. Further validation on diverse and larger datasets is necessary to ensure their predictive reliability and biological insights, advancing the field beyond current limitations.

In this study, we explored a practical approach to predict the risk of degenerative diseases by leveraging the relationship between aging and these diseases through DNA methylation data. We implemented an epigenetic clock based on a multi-scale convolutional neural network (MSCNN), which accurately predicts biological age by capturing both local and global features of the methylation data. By using Shapley additive explanations (SHAPs), we identified key CpG sites that significantly contribute to the age prediction model, providing insights into the most relevant methylation sites associated with aging. Then, we conducted a proteomics analysis based on protein–protein interaction networks, where the functional enrichment of subnetworks was investigated to confirm 196 marker CpG sites. Finally, a multilayer perceptron (MLP)-based degenerative disease risk prediction model (Mlp-DDR) was well trained and tested for nine classical degenerative diseases.

## 2. Results

### 2.1. Validation and Performance Enhancement with Selected CpG Sites

To validate the DNA methylation sites linked to degenerative diseases, further experiments were conducted. We reintegrated 196 CpG sites identified from key subnetworks into the MSCAP model and tested it on five disease datasets. Figure 1 shows the performance comparison between the original MSCAP model with 25,789 sites and the revised model with 196 sites. The updated model consistently outperformed the original, significantly enhancing the predictive accuracy for degenerative diseases. This confirms the relevance of these CpG sites in disease mechanisms, supporting further investigation into their potential for disease prediction.

### 2.2. Model Performance Analysis

This study developed Mlp-DDR, a degenerative disease risk prediction model, based on a diverse array of diseases: Alzheimer’s disease, Parkinson’s disease, breast cancer, colon cancer, lung cancer, liver cancer, kidney cancer, osteoporosis, and brain cancer. Four commonly used metrics were selected for assessment: AUC, ACC, precision, and recall, with the AUC being the primary metric for evaluating model performance. The predictive performance of Mlp-DDR across these diseases is detailed in Table 1. Notably, Mlp-DDR achieved the highest predictive accuracy for Alzheimer’s disease, with an AUC of 0.96, closely followed by Parkinson’s disease and breast cancer.

## 3. Discussions

### 3.1. Comparative Analysis of Machine Learning Models

In addition to the Mlp-DDR model, this study evaluated the performance of six other machine learning algorithms: K-nearest neighbors (KNN) [20], logistic regression (LR), naive Bayes (NB) [21], random forest (RF) [22], support vector machine (SVM) [23], and extreme gradient boosting (XGBoost) [24]. The predictive efficacy of these classifiers was assessed using five-fold cross-validation and four performance metrics. Table 2 provides a comparative overview of the predictive performance of these seven models across nine degenerative disease datasets, showcasing Mlp-DDR’s superior performance across all conditions assessed.

The establishment of the other six classification models involved an initial division of the training and test sets using five-fold cross-validation. Within the training sets, a grid search coupled with three-fold cross-validation was utilized to optimize the models’ parameters for the best possible outcomes. The final classification results were determined by averaging the outcomes of five experiments. Except for the Mlp-DDR model, the naive Bayes model performed the bets across four diseases, while the remaining models demonstrated good performance on two to three diseases. Mlp-DDR was particularly notable for its exceptional predictive accuracy in Alzheimer’s disease, demonstrating its efficacy with an AUC of 0.96, as further evidenced by the ROC curves shown in Figure 2.

### 3.2. Performance Variability among Different Diseases

The variability in the performance of the Mlp-DDR model across different degenerative diseases underscores the impact of dataset characteristics and disease complexity on predictive accuracy. The model achieved high predictive accuracy for Alzheimer’s disease, Parkinson’s disease, and breast cancer, which can be attributed to the availability of ample and high-quality data samples. These diseases had well-balanced datasets including both positive and negative samples, which helps avoid prediction biases caused by data imbalance. Additionally, the selection of DNA methylation sites closely related to these diseases allowed the model to better capture their unique characteristics and patterns, enhancing predictive reliability.

In contrast, the model exhibited lower accuracy in predicting liver cancer, lung cancer, kidney cancer, brain cancer, and osteoporosis. This reduced performance may be attributed to the inherent heterogeneity of these cancers, which includes varying subtypes, stages, and environmental factors that add layers of complexity to the disease profiles. Moreover, the datasets for these conditions were often smaller and less balanced compared to those for neurodegenerative diseases and breast cancer, posing significant challenges for the predictive model. These complexities may hinder the model’s ability to accurately discern underlying patterns, affecting its overall effectiveness in these conditions.

### 3.3. Insights from DNA Methylation Data Analysis

The Mlp-DDR model effectively captures the complex relationships within DNA methylation data through its neural network nodes, thereby achieving accurate predictions for degenerative diseases. The DNA methylation sites associated with these diseases enable the model to better learn and understand the diverse patterns of DNA methylation across different disease states. These insights are not only of practical significance for the early diagnosis of diseases but also provide a powerful tool for the in-depth exploration of the degenerative mechanisms of DNA methylation.

By identifying specific DNA methylation sites that are closely linked to disease mechanisms, the model can guide early intervention strategies, allowing for more personalized and timely treatment plans. Furthermore, understanding these methylation patterns could lead to the development of novel biomarkers, improving the precision of diagnostic tests and potentially leading to targeted therapies that address the underlying causes of degenerative diseases. By enhancing our understanding of disease pathogenesis, this approach offers potential for improving the methods used for disease treatment and management, ultimately contributing to better patient outcomes and quality of life.

### 3.4. Limitations and Future Work

Despite the significant progress made in developing the deep-learning-based epigenetic clock and disease risk prediction model, several limitations remain. The high capacity and complexity of the model, while improving accuracy, result in increased time complexity and demand for advanced experimental environments. Future work could focus on optimizing the model architecture and employing distributed training strategies to enhance efficiency.

Recent research highlights the importance of DNA methylation in bone and joint diseases, such as osteoporosis and osteoarthritis. For example, m6A modification and key loci identified through GWAS and multi-omic analyses demonstrate the critical role of DNA methylation in these conditions [7,8]. Incorporating these insights could extend our model’s applicability, offering more targeted strategies for treating a broader range of degenerative diseases.

Additionally, creating tissue-specific epigenetic clocks could improve model specificity and allow for more precise disease prediction within particular tissues. Expanding the dataset to include a wider range of diseases, such as tumors or adolescent diseases, would further enhance the model’s generalizability and applicability.

These efforts will enhance the model’s practicality and scalability, supporting its broader application in clinical settings and disease prediction tasks.

## 4. Materials and Methods

### 4.1. Datasets

DNA methylation data and clinical information of nine degenerative diseases were collected from the GEO database [25]: Alzheimer’s disease, Parkinson’s disease, breast cancer, colon cancer, lung cancer, liver cancer, kidney cancer, osteoporosis, and brain cancer. Only those samples with more than 20 individuals and no missing labels were included, resulting in a total of 2248 individual samples. The corresponding disease type distribution is shown in Figure 3. The involved diseases cover multiple organ systems such as the nervous system, skeletal system, liver, respiratory system, and digestive system, encompassing neurodegenerative diseases leading to memory loss, movement disorders, and cognitive decline, various cancers with progressive tissue damage, and skeletal system diseases.

Parkinson’s disease and Alzheimer’s disease are both neurodegenerative conditions affecting the nervous system, yet they have different stages of disease progression. In the early stages of Parkinson’s disease, patients may exhibit mild symptoms such as hand tremors, muscle stiffness, and reduced mobility. As the disease progresses to the intermediate stage, symptoms typically worsen, and patients may experience more severe motor impairments, including unstable gait, bradykinesia (slowness of movement), and increased muscle rigidity. In the advanced stages of Parkinson’s disease, severe motor disabilities can occur, potentially impacting every aspect of daily life. Patients at this stage may require constant care and support.

The staging of Alzheimer’s disease often utilizes clinical dementia ratings to assess the extent of cognitive impairment. In the early preclinical stage, patients may not show significant cognitive problems, although they might experience mild memory issues. As the disease progresses to mild cognitive impairment, patients may experience mild memory loss and a decline in cognitive function but are still capable of carrying out daily activities. In the moderate cognitive impairment stage, symptoms intensify, and patients may encounter more significant difficulties in performing daily tasks, often requiring assistance. In the later severe cognitive impairment or dementia stage, patients may completely lose the ability to care for themselves, with almost total loss of recognition of their environment and the people around them.

For the clinical data, only the “Pathological Staging” information of the patients was retained. Any sample lacking this information was removed. The staging information of Parkinson’s disease can be classified as “early Parkinson’s” and the staging information of Alzheimer’s disease can be classified as “asymptomatic” and “mild cognitive impairment” as the early stages of these degenerative diseases. Samples classified as “mid-stage Parkinson’s”, “moderate cognitive impairment”, “late-stage Parkinson’s”, and “severe cognitive impairment (dementia)” are defined as the late stages of these degenerative diseases. Table 3 presents the staging information for the nine degenerative diseases. In the clinical data labels, “0” represents the early stage and “1” represents the late stage.

### 4.2. Model Construction

The entire prediction process is composed of three major components: (1) an epigenetic clock model was used to identify DNA methylation sites highly correlated with age; (2) a biomics analysis module was used to screen the degenerative disease CpG marker sites; (3) a multilayer perceptron module was used to classify the nine types of degenerative diseases.

The epigenetic clock model was constructed based on a multi-scale convolutional neural network (MSCNN), termed MSCAP [26]. Since degenerative diseases are directly correlated with biological age, the MSCAP model, which accurately predicts age, can be used to identify DNA methylation sites highly correlated with age. Next, we utilized the SHAP tool, which calculates the contribution of DNA methylation sites to the model, for weight analysis of the MSCAP model. This analysis screened 1071 key DNA methylation sites critical for the model’s age prediction. Subsequently, we employed bioinformatics tools such as a protein interaction network analysis and KEGG pathway analysis to select 196 DNA methylation sites associated with degenerative diseases. Finally, a dataset containing 196 key DNA methylation sites from 2248 disease samples was fed into a multilayer perceptron module. This module processes the data through three hidden layers, each comprising 64 nodes. The model’s output consists of two labels: 0 (early-stage disease) and 1 (late-stage disease). Figure 4 illustrates the flowchart of the degenerative disease risk prediction model.

### 4.3. Epigenetic Clock Based on Deep Learning

The multi-scale convolutional age prediction (MSCAP) model employs a multi-scale convolutional neural network (MSCNN) architecture structured into four main components: data input, two fully connected layers, a multi-scale convolutional neural network module, and a final regression output layer. Initially, DNA methylation data from 25,789 CpG sites shared between the Illumina 27K [27] and 450K platforms [28] are input into the model. The first fully connected layer processes these inputs nonlinearly, using a sigmoid activation function to derive correlations between pairs of CpG sites. Subsequently, the MSCNN module, incorporating blocks with varying convolutional kernel sizes (1, 8, 16, and 32), processes the feature vectors to extract both local and global contextual features effectively. This design allows for a robust capture of complex interrelationships within the data, significantly enhancing predictive performance. The final output is generated through a second fully connected layer, which provides the predicted DNA methylation age based on the learned features.

The MSCAP model was trained using DNA methylation data from a diverse set of samples and compared with existing epigenetic clock models, consistently demonstrating enhanced prediction accuracy across multiple independent test datasets. This approach underscores the model’s capability to handle complex, high-dimensional datasets, thereby providing a solid foundation for the further development of predictive models for degenerative diseases.

### 4.4. Identification of Age-Related CpG Sites

Shapley additive explanations (SHAPs) [29,30], derived from the cooperative game theory, quantify the impact of each feature on a model’s output using Shapley values [31], providing a consistent and interpretable measure of feature importance. This analysis was utilized to determine the contribution of individual CpG sites to the predictive accuracy of the MSCAP model. By analyzing SHAP values, 1071 key CpG sites were identified based on their substantial influence. These sites were then mapped to their corresponding genes using genome annotation databases like the UCSC Genome Browser [32] and Ensembl [33], as well as bioinformatics tools such as BEDTools [34].

This mapping process involved matching coordinates and resulted in these CpG sites being associated with 1033 distinct genes, with some CpG sites mapping to the same gene. Table 4 summarizes part of the mapping results, listing CpG IDs along with their SHAP values, associated gene names, and chromosomal positions. This mapping is crucial for understanding how these CpG sites contribute to age prediction and their potential roles in biological pathways related to aging and degenerative diseases. The selected CpG sites enable a deeper biological interpretation of the MSCAP model’s outcomes, thereby enhancing the understanding of DNA methylation patterns associated with age prediction.

### 4.5. Enrichment Analysis

We leveraged the STRING database [35] and Cytoscape platform [36] to analyze protein interactions and construct biological networks [37,38], aiming to identify key molecular interactions related to degenerative diseases. STRING, integrating data from comprehensive sources including UniProt and KEGG [39], enabled the mapping of a vast protein interaction network using 1033 genes linked to significant CpG sites [40,41]. We then employed Cytoscape, enhanced with the Molecular Complex Detection (MCODE) plugin, to delve deeper into this network. This analysis identified densely connected regions, or subnetworks, which were prioritized based on their connectivity scores. Ultimately, the top four subnetworks with the highest scores were selected as key subnetworks, and the top four highest-scored subnetworks are depicted in Figure 5.

We subsequently utilized the Metascape platform [42] for a functional enrichment analysis [43,44] of the genes within these critical subnetworks, setting the significance threshold at a *p*-value of 0.05. The second-highest-scoring subnetwork predominantly involves pathways related to degenerative diseases, suggesting that the genes it includes could play crucial roles in regulating biological processes associated with these diseases. We consider the genes in the Top2 subnetwork as key to understanding the pathogenesis of degenerative diseases. By linking these genes to their corresponding DNA methylation sites, we identified key CpG sites associated with degenerative diseases, involving 83 genes corresponding to 196 DNA methylation sites. Figure 6 displays the KEGG pathway enrichment analysis heatmap for the Top2 subnetwork.

### 4.6. Hyperparameter Tuning and Evaluation

The optimization of model hyperparameters is principally aimed at identifying the set of hyperparameters that enables the algorithm to achieve optimal performance on a verifiable dataset. However, manual tuning often depends on the researcher’s practical experience and is time consuming and labor intensive; thus, automated hyperparameter optimization is indispensable. This paper employs a grid search approach for hyperparameter tuning coupled with a five-fold cross-validation method to evaluate the optimized model’s classification effectiveness.

Model optimization is a critical step in ensuring the best performance of a model. To select the optimal neural network architecture, this study employs grid search to systematically explore the impact of multiple key hyperparameters on the model’s outcomes, including the number of hidden layers (2, 5, or 8), the number of neurons in each hidden layer (16, 32, or 48), the dropout rate (0 or 0.1), the application of the batch normalization technique, regularization methods (L1 regularization, L2 regularization, or Elastic Net regularization), and the choice of activation functions (ReLU or SeLU). Additionally, this study considers the setting of the learning rate, experimenting with three different values: 0.0002, 0.0005, and 0.001. To ensure the stability and validity of the experiments, certain parameters are fixed, including the use of the Adam optimizer, a batch size set to 128, a fixed training period of 200 epochs, and the use of binary cross-entropy as the loss function. A learning rate decay mechanism is also introduced; if the training loss remains stable for 30 consecutive epochs, the learning rate is decayed by a factor of 0.2 to promote model convergence.

The grid search was utilized to exhaustively search through all possible hyperparameter combinations, training multiple neural network models and evaluating each model’s performance through five-fold cross-validation. The dataset was divided into five exclusive subsets. In each round of cross-validation, one subset is chosen as the validation set, while the remaining four serve as the training set, and the Area Under the ROC Curve (AUC) is used as the performance metric. For each validation set, the model’s AUC value is computed as the performance measure. Finally, the average AUC value obtained from the five cross-validation folds is taken as the ultimate metric for assessing model performance. By employing five-fold cross-validation, the model’s generalization ability and stability are more comprehensively evaluated, ensuring the selected model’s adaptability to unknown data.

After the grid search and cross-validation, the best-performing hyperparameter combination on the validation set was selected. The network consists of three hidden layers, each with 64 neurons, without dropout, and includes batch normalization. The SeLU activation function and Elastic Net regularization were applied, with the learning rate set at 0.001. The model displayed good fitting ability on the training set and also demonstrated high classification accuracy on the test set.

## 5. Conclusions

In this study, we developed and validated Mlp-DDR, a multi-layer-perceptron-based model for predicting the risk of nine degenerative diseases using DNA methylation data. Our approach involved constructing an epigenetic clock based on a multi-scale convolutional neural network to identify key CpG sites associated with aging, followed by the selection of 196 marker CpG sites through a proteomics and bioinformatics analysis. These selected CpG sites were then used to train the Mlp-DDR model for accurate disease risk prediction.

The Mlp-DDR model demonstrated superior predictive performance across the nine disease types, achieving the highest accuracy for Alzheimer’s disease with an AUC of 0.96. This model outperformed six other machine learning algorithms (K-nearest neighbors, logistic regression, naive Bayes, random forest, support vector machine, and extreme gradient boosting) in terms of key performance metrics such as AUC, accuracy, precision, and recall. The results underscore the potential of deep learning models in handling complex, high-dimensional biological data, thereby offering a significant advancement in the early prediction of degenerative diseases.

In future works, efforts should be focused on enhancing the interpretability of these models and validating their performance across larger and more diverse datasets. Additionally, exploring the integration of multi-omics data and longitudinal studies could further improve the predictive power and clinical applicability of these models, ultimately contributing to better disease management and patient outcomes.

## Figures and Tables

**Figure 1 ijms-25-09514-f001:**
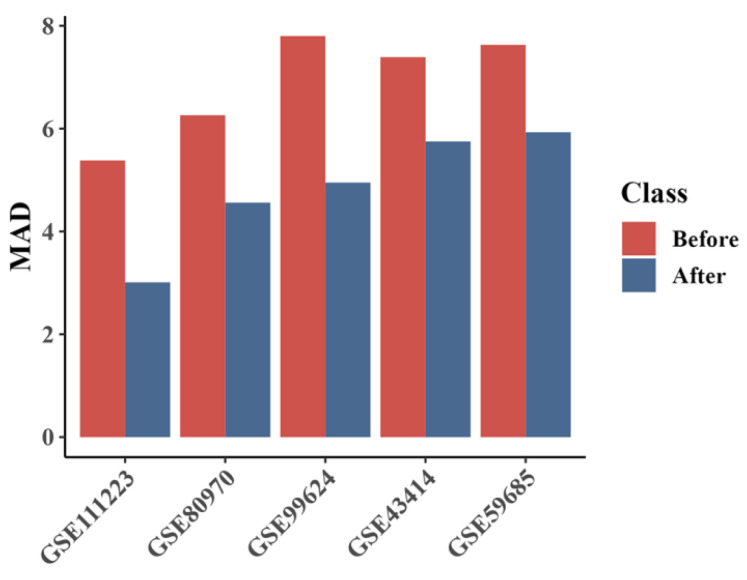
Comparison of the results before and after retraining the MSCAP model using selected CpG sites. This figure displays the performance comparison of the MSCAP model using the original 25,789 sites (‘Before’) and after retraining with the 196 selected methylation sites (‘After’) across five degenerative disease datasets: GSE111223, GSE80970, GSE99624, GSE43414, and GSE59685. The graph shows that the retrained model with fewer sites significantly outperforms the original, indicating that these sites are highly predictive of degenerative diseases. The Mean Absolute Deviation (MAD) scores are lower in the ‘After’ scenario, demonstrating improved model accuracy.

**Figure 2 ijms-25-09514-f002:**
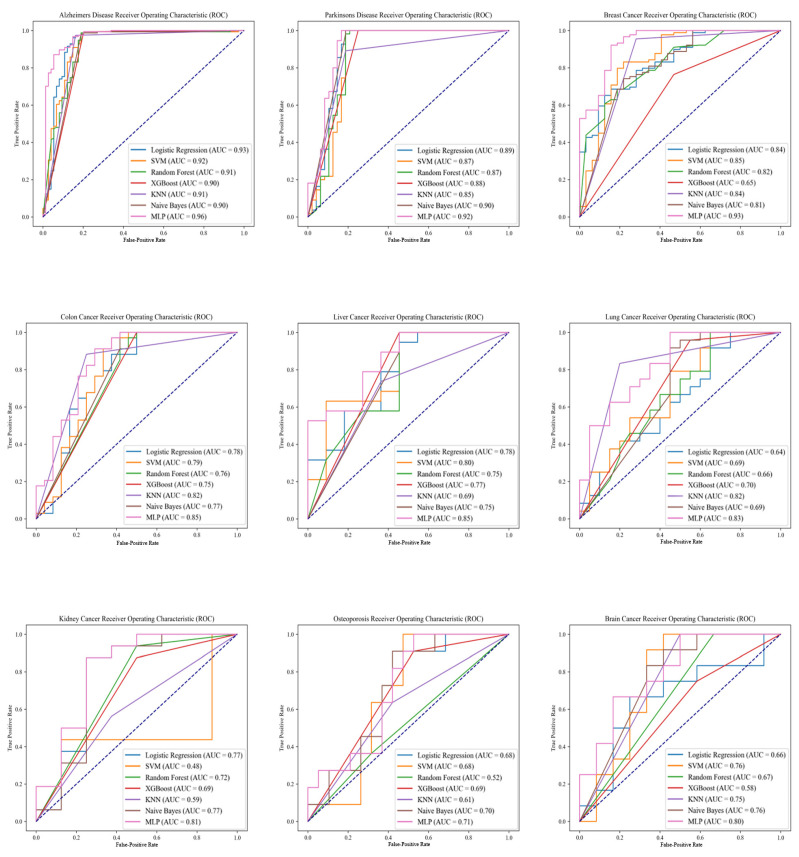
ROC curves of seven models for nine degenerative diseases. This figure illustrates the ROC curves for various models tested across nine degenerative diseases: Alzheimer’s disease, Parkinson’s disease, breast cancer, colon cancer, lung cancer, liver cancer, kidney cancer, osteoporosis, and brain cancer. Each curve represents the performance of a different predictive model (MLP, logistic regression, naive Bayes, random forest, SVM, and XGBoost) across these diseases. The Area Under the Curve (AUC) for each model is indicated in the legend, reflecting the models’ ability to distinguish between disease and non-disease states effectively. The Alzheimer’s disease model, particularly MLP-DDR, achieved the highest AUC of 0.96, demonstrating its superior predictive accuracy. The data used for this analysis were collected from the GEO database, encompassing DNA methylation data and clinical information for the nine degenerative diseases.

**Figure 3 ijms-25-09514-f003:**
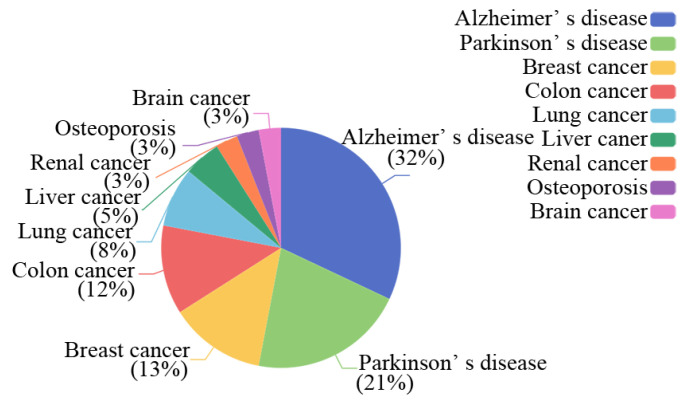
Pie chart of the dataset distribution of degenerative diseases. This pie chart clearly delineates the distribution of the dataset across nine different types of degenerative diseases. Alzheimer’s disease comprises the largest segment at 32%, followed by Parkinson’s disease at 21%, emphasizing their prominent representation in the dataset. Both breast cancer and colon cancer are significant portions of the dataset, with breast cancer accounting for 13% and colon cancer for 12%. The remaining five diseases are sorted in order of their share. The data used for this analysis were sourced from the GEO database.

**Figure 4 ijms-25-09514-f004:**
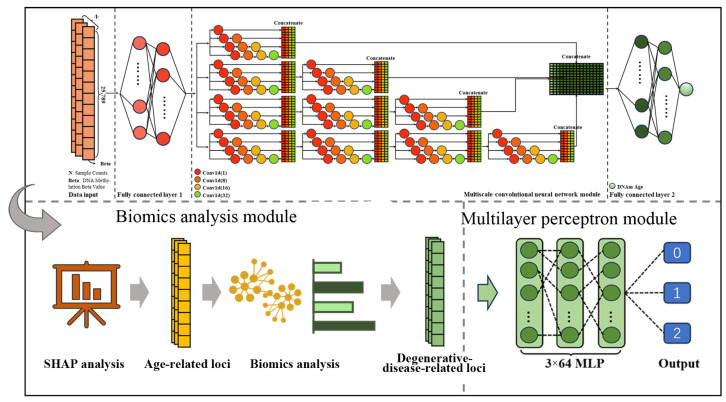
Flow chart of risk prediction for degenerative diseases. This figure illustrates the structured flowchart of the degenerative disease risk prediction model, divided into three phases. First, an epigenetic clock model, MSCAP, is constructed using a multi-scale convolutional neural network (MSCNN) to identify DNA methylation sites with significant degenerative patterns. Next, the SHAP tool analyzes the MSCAP model’s weights, identifying 1071 significant DNA methylation sites. Through proteomics analysis, 196 sites associated with degenerative disease mechanisms are selected. Finally, 2248 disease samples, including the 196 key sites, are input into a multilayer perceptron (MLP) module with three hidden layers of 64 nodes each, outputting two labels: 0 (early-stage disease) and 1 (late-stage disease).

**Figure 5 ijms-25-09514-f005:**
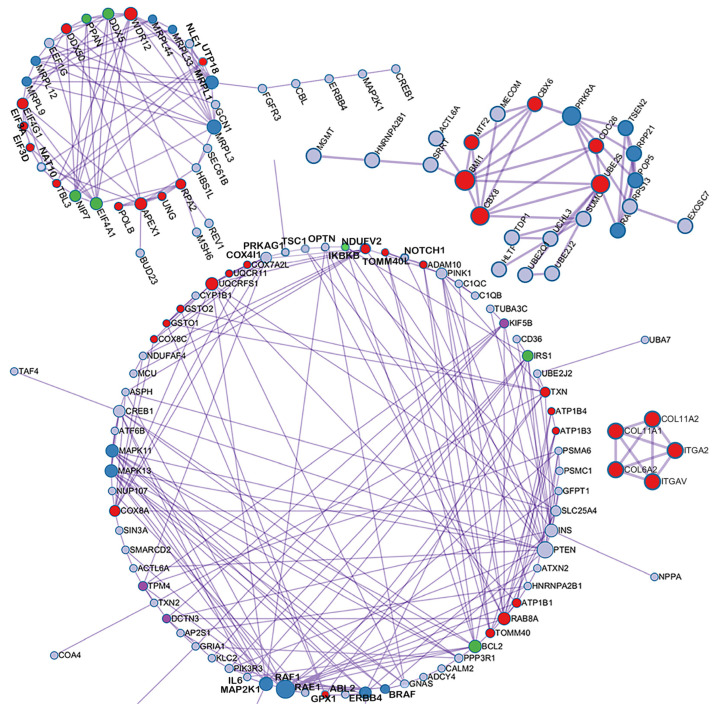
Four key subnetworks. This figure displays the top four subnetworks identified using STRING and Cytoscape with the MCODE plugin, which highlight crucial gene interactions linked to degenerative diseases. Node size and color indicate the gene’s importance and connectivity; larger and darker nodes signify higher significance.

**Figure 6 ijms-25-09514-f006:**
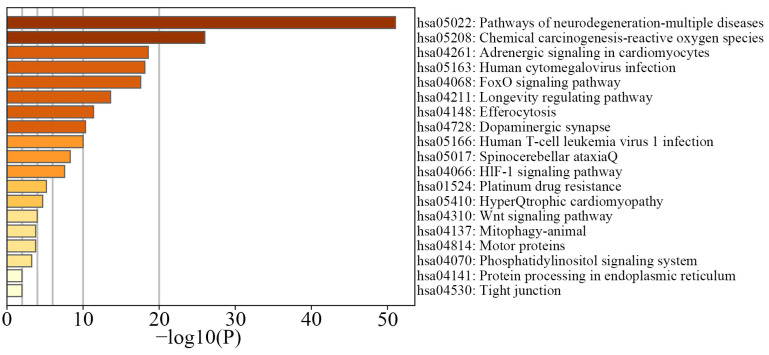
KEGG pathway analysis of the Top2 subnetwork. This figure displays a heatmap of the KEGG pathway enrichment analysis for the Top2 subnetwork, derived from our research using the Metascape platform. Each bar in the heatmap represents a specific pathway, with the color intensity reflecting the −log10(P) value, which quantifies the statistical significance of the enrichment. Higher values, indicated by darker colors, suggest stronger associations between these pathways and the genes within the Top2 subnetwork, underscoring their potential impact on degenerative disease mechanisms.

**Table 1 ijms-25-09514-t001:** Predictive performance of Mlp-DDR on nine degenerative diseases.

Degenerative Disease Type	AUC	ACC	Recall	Precision
Alzheimer’s Disease	0.96	0.92	0.97	0.92
Parkinson’s Disease	0.92	0.90	0.96	0.87
Breast Cancer	0.93	0.91	0.97	0.91
Colon Cancer	0.85	0.83	0.97	0.79
Lung Cancer	0.85	0.77	0.89	0.77
Liver Cancer	0.83	0.73	0.62	0.83
Kidney Cancer	0.81	0.83	0.94	0.83
Osteoporosis	0.71	0.63	0.82	0.50
Brain Cancer	0.80	0.71	1.00	0.63

**Table 2 ijms-25-09514-t002:** (**a**) Comparative predictive performance of seven models on nine degenerative disease datasets. (**b**) Comparative predictive performance of seven models across nine degenerative disease datasets. (**c**) Comparative predictive performance of seven models across nine degenerative disease datasets.

(**a**)					
**Disease Type**	**Model**	**AUC**	**ACC**	**Precision**	**Recall**
Alzheimer’s Disease	MLP	**0.96**	0.92	0.97	0.92
	LR	0.93	0.93	0.99	0.91
	SVM	0.92	0.93	0.99	0.91
	RF	0.91	0.93	0.99	0.91
	XGBoost	0.90	0.93	0.99	0.91
	KNN	0.91	0.93	0.97	0.93
	NB	0.90	0.93	0.98	0.92
Parkinson’s Disease	MLP	**0.92**	0.90	0.96	0.87
	LR	0.89	0.90	0.98	0.86
	SVM	0.87	0.91	1.00	0.86
	RF	0.87	0.90	0.98	0.86
	XGBoost	0.88	0.88	1.00	0.82
	KNN	0.85	0.85	0.89	0.84
	NB	0.90	0.91	1.00	0.86
Breast Cancer	MLP	**0.93**	0.91	0.97	0.91
	LR	0.84	0.79	1.00	0.77
	SVM	0.85	0.86	1.00	0.84
	RF	0.82	0.74	0.79	0.85
(**b**)					
**Disease Type**	**Model**	**AUC**	**ACC**	**Precision**	**Recall**
Breast Cancer	XGBoost	0.65	0.70	0.76	0.82
	KNN	0.84	0.89	0.96	0.90
	NB	0.81	0.72	0.69	0.91
Colon Cancer	MLP	**0.85**	0.83	0.97	0.79
	LR	0.78	0.59	1.00	0.59
	SVM	0.79	0.59	1.00	0.59
	RF	0.76	0.79	1.00	0.74
	XGBoost	0.75	0.79	1.00	0.74
	KNN	0.82	0.83	0.88	0.83
	NB	0.77	0.78	0.91	0.76
Lung Cancer	MLP	**0.85**	0.77	0.89	0.77
	LR	0.78	0.63	1.00	0.63
	SVM	0.80	0.77	0.89	0.77
	RF	0.75	0.83	1.00	0.79
	XGBoost	0.77	0.83	1.00	0.79
	KNN	0.69	0.70	0.74	0.78
	NB	0.75	0.83	1.00	0.79
Liver Cancer	MLP	**0.83**	0.73	0.62	0.83
	LR	0.64	0.55	1.00	0.55
	SVM	0.69	0.64	0.79	0.63
	RF	0.66	0.61	0.71	0.63
	XGBoost	0.70	0.73	0.96	0.68
	KNN	0.82	0.82	0.83	0.83
	NB	0.69	0.70	0.83	0.69
Kidney Cancer	MLP	**0.81**	0.83	0.94	0.83
	LR	0.77	0.67	1.00	0.67
	SVM	0.48	0.67	1.00	0.67
	RF	0.72	0.79	0.94	0.79
	XGBoost	0.69	0.75	0.88	0.78
	KNN	0.59	0.58	0.56	0.75
	NB	0.77	0.79	0.94	0.79
Osteoporosis	MLP	**0.71**	0.63	0.82	0.50
	LR	0.68	0.37	1.00	0.37
	SVM	0.68	0.37	1.00	0.37
	RF	0.52	0.57	0.36	0.40
	XGBoost	0.69	0.63	0.91	0.50
	KNN	0.61	0.60	0.64	0.47
	NB	0.70	0.60	1.00	0.48
(**c**)					
**Disease Type**	**Model**	**AUC**	**ACC**	**Precision**	**Recall**
Brain Cancer	MLP	**0.80**	0.71	1.00	0.63
	LR	0.66	0.50	1.00	0.50
	SVM	0.76	0.50	1.00	0.50
	RF	0.67	0.67	1.00	0.60
	XGBoost	0.58	0.58	0.75	0.56
	KNN	0.75	0.75	1.00	0.67
	NB	0.76	0.75	0.83	0.71

**Table 3 ijms-25-09514-t003:** Staging information of degenerative diseases.

Disease	Stage	Sample Count
Alzheimer’s Disease	Early	401
	Late	321
Parkinson’s Disease	Early	229
	Late	238
Breast Cancer	Early	148
	Late	134
Colon Cancer	Early	146
	Late	121
Lung Cancer	Early	84
	Late	104
Liver Cancer	Early	68
	Late	46
Kidney Cancer	Early	37
	Late	39
Osteoporosis	Early	38
	Late	34
Brain Cancer	Early	33
	Late	27

**Table 4 ijms-25-09514-t004:** Coordinate information of partial CpG sites.

CpG ID	SHAP Values	Gene Name	Chromosome Position
cg00008493	−3.31 × 10^−5^	COX8C	14
cg00024812	−5.67 × 10^−5^	CPSF3	2
cg00033773	−4.54 × 10^−5^	MORG1	19
cg00094319	−2.96 × 10^−5^	PABPC3	13
cg00103783	−8.55 × 10^−5^	MPDU1	17
cg00136477	−7.97 × 10^−5^	C1QC	1
cg00187686	−2.08 × 10^−5^	TCN1	11
cg00234961	−2.22 × 10^−5^	ZBED4	22

## Data Availability

Data are contained within the article or Appendix A.

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
