# Peer review of "Prediction of Multiple Degenerative Diseases Based on DNA Methylation in a Co-Physiology Mechanisms Perspective"

_ijms, 2024, doi:10.3390/ijms25179514_

Round 1
Reviewer 1 Report
Comments and Suggestions for Authors
Major reorganization of the paper is needed to enhance readability.
The labels in figure 6 are difficult to read. Please increase the font size.
The authors have figures and a table in the methods section which should be part of results section.
The authors should separate the results and dicussions and in detail discuss the findings of their study and how it compares to other studies. They should also mention how the findings of their study potentially benefit patient diagnosis or treatment.
Additionally, they should have a separate paragraph in the discussion section detailing the limitations of their study and the model.
Comments on the Quality of English Language
minor grammar editing needed.
Reviewer 2 Report
Comments and Suggestions for Authors
This paper entitled “Multiple Degenerative Diseases Prediction Based on DNA Methylation in a Co-Physiology-Mechanisms Perspective” is presented. The study is potentially interesting. The methods used were appropriate and the conclusions were reasonable.
Some possible issues?
Figure 1. Pie chart of data set distribution of degenerative diseases. Can you state the source of data analyses in the figure legends?
Figure 5. Comparison of results before and after retraining the MSCAP model. Can you state the sample size of analyses? N=? with comparative p Values?
Figure 6. ROC Curves of Seven Models for Nine Degenerative Diseases. Can you state the source of data analyses in the figure legends?
Recent studies have found that Methylation regulates bone and joint diseases (for example, PMID: 34993198, PMID: 38041181). It would be relevant to discuss include or discuss DNA Methylation in bone and joint diseases as reported.
Round 2
Reviewer 1 Report
Comments and Suggestions for Authors
Thanks to the authors for making the required edits.